# On stabilisation of compositional density jumps in compressible mantle convection simulations

Paul James Tackley<sup>1</sup>

<sup>1</sup>Department of Earth and Planetary Sciences, ETH Zurich, Zurich, 8092, Switzerland

5 Correspondence to: Paul J. Tacklev (ptacklev@ethz.ch)

Abstract. Large density jumps in numerical simulations of solid Earth dynamics can cause numerical "drunken sailor" oscillations. An implicit method has previously been shown to be very effective in stabilising the density jump that occurs at a free surface against such instabilities (Kaus et al., 2010; Duretz et al., 2011). Here the use of this to prevent oscillations of compositional layers deeper in the mantle is examined. If the stabilisation algorithm uses the total density field including the steady increase of density with depth due to adiabatic compression and jumps due to phase transitions then a severe artificial reduction of convective vigour occurs because the algorithm assumes that density is advected with the flow but these density gradients are not. This artificial vigour reduction increases with Rayleigh number but decreases with decreasing grid spacing. Thus, it is essential to use only composition-related density gradients in the stabilisation algorithm, and a simple method for isolating these is presented. Once this is done, the stabilisation method works effectively for internal compositional layers as well as a free surface.

# 1. Introduction

15

Density jumps due to treatment of a free surface by the "sticky air" method (e.g. Crameri et al., 2012), in which the surface is represented as an abrupt interface between rock and low-density, low-viscosity "air", can induce numerical "drunken sailor" instabilities, in which the free surface oscillates up and down on successive time-steps, overshooting its equilibrium position. To cure this problem, an implicit free-surface stabilisation algorithm was introduced, initially for the finite-element discretization (Kaus et al., 2010; Andrés-Martínez et al., 2015) and then for the staggered-grid finite difference (equivalent to finite-volume) discretization (Duretz et al., 2011). The basis of this scheme is that when calculating the flow field, the change in buoyancy due to advection of density during a time step is treated implicitly; while this is applied throughout the domain, by far the largest correction comes from advection of the free surface. It is very effective in stabilising the free surface, allowing a normal (e.g. Courant condition limited) time-step to be used (Kaus et al., 2010; Duretz et al., 2011). A subsequent rigorous stability analysis led to an alternative approach using an explicit scheme based on nonstandard finite differences (Rose et al., 2017). Fully implicit time stepping in the entire domain is another alternative (Popov and Sobolev, 2008; Kramer et al., 2012).

In global mantle convection simulations, compositional density jumps can also arise inside in the mantle, typically due to a primordial layer of dense material above the core-mantle boundary (CMB) (e.g. Gurnis, 1986; Tackley, 1998; Davaille, 1999; Deschamps et al., 2011) or a layer of dense subducted basaltic crust above the CMB (e.g. Christensen and Hoffmann, 1994; Ogawa, 2000; Nakagawa and Tackley, 2015), and although the associated density difference is much smaller than that at a free surface, it can sometimes be enough to also induce numerical "drunken sailor" oscillations. Thus, it is tempting to apply the implicit density jump stabilisation (DJS) algorithm throughout the domain. However, a key assumption of the algorithm is that density is advected with the flow, but this is not the case for the steady density increase with depth (pressure) due to adiabatic compression or density jumps due to solid-solid phase transitions, the most important of which cause the 410 km and 660 km seismic discontinuities. According to the most commonly-used Earth model PREM (Preliminary Reference Earth Model) (Dziewonski and Anderson, 1981), the density jump at 660 km is about 10%, while the density increase due to combined compression and phase transitions from the surface to the CMB is about 65%. If the DJS algorithm is applied to these density gradients then a substantial artificial reduction in convective vigour results, as shown later. Thus, it is important to apply the DJS algorithm only to density gradients or jumps due to compositional gradients and not density gradients/jumps due to adiabatic compression or phase transitions.

In this paper, a test code written in the Julia programming language (Bezanson et al., 2017) is used first to again demonstrate the effectiveness of the DJS algorithm for stabilising a free surface, to additionally show its effectiveness in preventing oscillations of a dense layer above the CMB, then to quantify the artificial reduction in convective vigour when applying it to total density in a setup that includes adiabatic compression and/or a phase transition. A way of separating composition-related density gradients from the total density gradient for arbitrary density functions is then presented.

# 2. Mathematical background

# 2.1 Momentum equation with density jump stabilisation

Here the basics of the algorithm are reviewed, following Duretz et al., (2011). The equation of motion (force balance) for highly viscous flow in Earth's mantle and crust is the Stokes equations, which neglect inertial terms in the Navier-Stokes equations.

$$-\nabla p + \nabla \cdot \underline{\underline{\tau}} = -\rho \, \vec{g} \tag{1}$$

where p is pressure,  $\rho$  is density,  $\vec{g}$  is gravity and  $\underline{\underline{\tau}}$  is the deviatoric stress tensor, given by:

$$\tau_{ij} = \eta \left( \frac{\partial v_i}{\partial x_j} + \frac{\partial v_j}{\partial x_i} - \delta_{ij} \frac{2}{n} \nabla \cdot \vec{v} \right). \tag{2}$$

 $\eta$  is the dynamic viscosity and n is the number of spatial dimensions (2 or 3). For incompressible flow, the last term is zero.

During a time step, advection of composition in the vicinity of a density gradient/jump can substantially change the density on the right-hand-side of equation (1), approximately as:

$$\rho_{new} = \rho - \Delta t (\vec{v} \cdot \nabla \rho) \tag{3}$$

where  $\Delta t$  is the time step. These density changes can be treated implicitly by substituting  $\rho_{new}$  for  $\rho$  in (1) and moving the velocity term to the left-hand side:

$$-\nabla p + \nabla \cdot \underline{\tau} - \theta \Delta t (\vec{v} \cdot \nabla \rho) \vec{g} = -\rho \vec{g} \tag{4}$$

where  $\theta$  is a factor between 0 (explicit) and 1 (implicit). The finite-difference stencil for velocity components is modified accordingly, based on  $\nabla \rho$  calculated at the beginning of the time step. In practice, in the vicinity of a near-horizontal layer interface it is vertical motions that change the density so a simplified version considering only vertical (z) velocities and assuming that g is vertical has almost the same stabilisation effect:

$$-\nabla p + \nabla \cdot \underline{\underline{\tau}} - \theta \Delta t \left( v_z \frac{\partial \rho}{\partial z} \right) g \hat{\overline{z}} = -\rho g \hat{\overline{z}}$$
 (5)

#### 2.2 Continuity equation

The full continuity (conservation of mass) equation can be written in Eulerian form as

$$\nabla \cdot (\rho \vec{v}) = -\frac{\partial \rho}{\partial t} \tag{6}$$

or Lagrangian form as

$$\rho \nabla \cdot \vec{v} = -\frac{D\rho}{Dt} \tag{7}$$

These equations are commonly approximated bearing in mind that thermally induced density differences are of order 1%, which is much smaller than density differences due to adiabatic compression + phase transitions over the depth of the mantle (~65%) or due to compositional differences such as a free surface (discussed above) or iron diapirs (e.g. Samuel and Tackley, 2008; Lin et al., 2011) (~100%). Furthermore, for whole-mantle studies dynamic (i.e. related to the flow) pressure is much smaller than hydrostatic pressure, so its effect on density is typically ignored. Thus, (7) is often approximated as

$$\nabla \cdot \vec{v} = 0 \tag{8}$$

This is valid when density is advected with the flow, but invalid when there are significant non-advected density variations such as due to pressure or phase transitions. In this study it is necessary to model flow with both large pressure-related density variations and large composition-related density variations so a modified form of (7) is considered, decomposing the Lagrangian density time-derivative into temperature (T)-, pressure (P)- and composition (C)-related components:

$$\rho \nabla \cdot \vec{v} = -\frac{D\rho}{Dt} = -\left(\frac{D\rho}{Dt}\right)_T - \left(\frac{D\rho}{Dt}\right)_P - \left(\frac{D\rho}{Dt}\right)_C \tag{9}$$

T-induced variations are assumed to be negligible, consistent with the Boussinesq or compressible anelastic approximations

(additionally, T changes slowly in the Lagrangian frame). The compositional component is zero in the Lagrangian frame and, ignoring dynamic pressure as in the Boussinesq or anelastic approximation, the pressure component is due only to vertical motion as:

$$\rho \nabla \cdot \vec{v} = -\left(\frac{D\rho}{Dt}\right)_{p} = -v_{z} \frac{\partial \rho}{\partial z} \tag{10}$$

This can be satisfied using a z-dependent density that increases due only to hydrostatic compression:

$$\nabla \cdot (\rho_z \vec{v}) = 0 \tag{11}$$

where  $\rho_z$  is a depth-dependent reference density that depends only on hydrostatic compression, not composition. It would be possible to implement a more accurate version of the continuity equation that includes temperature and dynamic pressure effects (e.g. Gassmöller et al., 2020) but the current level of approximation suffices for the tests presented and is consistent with the commonly-used anelastic approximation (King et al. 2010).

#### 95 **2.3 Energy equation**

For the tests performed here a simple form of energy conservation is assumed:

$$\rho C_p \frac{\partial T}{\partial t} = k \nabla^2 T - \rho C_p \vec{v} \cdot \nabla T \tag{12}$$

where T is temperature, t is time,  $C_p$  is specific heat capacity, and k is thermal conductivity. Normally, when taking compressibility into account, terms for adiabatic heating/cooling and viscous dissipation would appear in this equation. This version corresponds to the limit of zero Grüneisen parameter ( $\gamma = \partial \ln T / \partial \ln \rho$ ), or in nondimensional terms, having a zero dissipation-number but finite compressibility-number (Tackley, 1996). The concept is to make the test program as simple as possible to demonstrate what is discussed in this manuscript.

#### 2.4 Test program


The associated test program CConv2dDJS.jl posted on Zenodo (Tackley and ETH Zurich, 2025) is written in the Julia programming language (Bezanson et al., 2017) and solves a nondimensional version of the equations above in two dimensions, x (horizontal) and z (vertical). The continuity equation is identical to equation (11) above, with  $\rho_z$  being 1.0 at the surface and increasing linearly to a specified value at the CMB. The two components of the momentum equation are:

$$-\frac{\partial p}{\partial x} + \frac{\partial \tau_{xx}}{\partial x} + \frac{\partial \tau_{xz}}{\partial z} = 0 \qquad -\frac{\partial p}{\partial z} + \frac{\partial \tau_{zz}}{\partial z} + \frac{\partial \tau_{zx}}{\partial x} - v_z \Delta t \frac{\partial \rho}{\partial z} B_{air} Ra = -Ra \left( T - B_{air} C_{air} - B_{DL} C_{DL} \right)$$
(13)

where Ra is the Rayleigh number,  $C_{\text{air}}$  and  $C_{\text{DL}}$  are the fraction (0-1) of air and dense layer, respectively, respectively, and  $B_{\text{air}}$  and  $B_{\text{DL}}$  are the compositional buoyancy ratios for air and dense layer, respectively ( $B_X = \Delta \rho_X / (\rho \alpha \Delta T)$ ) where  $\alpha$  is thermal expansivity and  $\Delta T$  is the temperature drop across the layer).  $\theta$  in equation (5) is assumed to be 1.0. For air, the buoyancy parameter is negative (air being less dense than rock) and has a value  $B_{air} = -1/(\alpha \Delta T) \approx -1/(2.10^{-5} \times 2500) = -20$  whereas  $B_{\text{DL}}$  is positive and has a much smaller magnitude. The mechanical boundary conditions are impermeable and free slip (zero shear stress) on all boundaries.

The nondimensional energy equation is:

$$\rho_{tot} \frac{\partial T}{\partial t} = \nabla^2 T - \vec{v} \cdot \nabla T \tag{14}$$

Thermal boundary conditions are insulating side boundaries and isothermal top and bottom boundaries (T=0 and 1, respectively). The equivalent equation for composition lacks the diffusion term.

Density is the sum of pressure-related, composition-related and phase transition-related components:

$$\rho_{tot} = \rho_z + \Delta \rho_C + \Delta \rho_{PT}$$
 (15)

Where  $\rho_z$  is 1.0 at the surface and increases linearly to a specified value at the CMB, the compositional component is given by

$$\Delta \rho_C = C_{air} \Delta \rho_{air} + C_{DL} \Delta \rho_{DL} = -C_{air} - C_{DL} B_{DL} / B_{air}$$
(16)

(noting that  $\Delta \rho_{air} = -1$  nondimensional and  $\Delta \rho_{DL} = \Delta \rho_{air} B_{DL} / B_{air} = -B_{DL} / B_{air}$ ), and  $\Delta \rho_{PT}$  is zero above the phase transition depth and the specified density increase below the phase transition depth, corresponding to a sharp phase transition with zero Clapeyron slope.

The continuity equation uses  $\rho_z$  as explained above, while the energy equation uses  $\rho_{tot}$ . The DJS algorithm can either correctly use  $\Delta\rho_C$  or incorrectly use  $\rho_{tot}$  in order to illustrate the bad artefacts that result.

The equations are discretized using a standard staggered-grid finite volume discretization (e.g. Harlow and Welch 130 1965; Patankar, 1980), as used by many codes in the geodynamical modelling community (e.g. Ogawa et al., 1991; Tackley, 1993; Trompert and Hansen, 1996; Gerya and Yuen, 2007; Kameyama et al. 2008; Tackley, 2008; Kaus et al., 2016). The velocity-pressure solution is solved with a direct solver utilising the built-in "\" operator. Advection of temperature and composition is performed using an upwind donor-cell technique, which is very diffusive but suffices for the tests here. Temperature diffusion is calculated using explicit finite differences.

# 135 **3. Results**





# 3.1 Surface or dense layer stabilisation

First, it is verified that the implementation of the DJS algorithm in the attached program eliminates "drunken sailor" oscillations. Figure 1 shows the effectiveness of the algorithm for preventing oscillations of a free surface with a sticky air layer. Detailed tests are not performed here because they have been already been reported elsewhere (Duretz et al., 2011). Figure 2 shows the effectiveness of the algorithm for stabilising a dense layer above the CMB. In both cases, oscillations occur with the algorithm switched off, but a smooth evolution is obtained with the algorithm switched on. When oscillations occur, the compositional interface (free surface or top of layer) becomes smeared out due to the numerical diffusion inherent in the upwind donor cell advection algorithm. In contrast, when the interface barely moves there is negligible numerical diffusion so the interface remains fairly sharp.

Figure 1. Stabilisation of a sticky-air layer with viscosity contrast 0.001 and thickness 0.1 on a 32x32 grid with Ra=10<sup>5</sup>. Oscillations occur when density jump stabilisation is switched off (left) but not when it is switched on (right).

# 3.2 Convection with depth-dependent density

Steady-state convection solutions are calculated for various density increases with depth ( $\rho_{cmb}/\rho_{surf}$ ), various Rayleigh numbers from  $10^4$  to  $3x10^5$ , and two grid resolutions (32x32 and 64x64). The calculations are run until the top and bottom Nusselt numbers are identical and the rms velocity has stopped changing, which typically requires 1000s of time steps. These tests do not have any compositional density variations so the DJS algorithm is not needed; their purpose is to demonstrate the problems that occur when it is applied to non-compositional density variations.

The influence of compressibility is tested first, varying the density increase with depth ( $\rho_{cmb}/\rho_{surf}$ ) from factor 1.0 to 2.0, bearing in mind that in Earth this ratio is about 1.65 (including compressibility and phase transitions). Correct solutions (using only non-existent composition-dependent density gradients in the DJS algorithm) are compared to those using the full density field (Figure 3 left column). Solutions indicate that the correct Nusselt number (top left) and  $V_{rms}$  (middle left) change slightly with  $\rho_{cmb}/\rho_{surf}$ , slightly increasing and decreasing, respectively. Resolution makes little difference to the

correct values. With DJS using the full density field, however, Nusselt number and  $V_{rms}$  decrease substantially as  $\rho_{cmb}/\rho_{surf}$  is increased. Ratios are plotted in the lower left. In the worst case ( $\rho_{cmb}/\rho_{surf}$ =2.0, 32x32 grid), Nusselt number is decreased by 35% and  $V_{rms}$  by about 65%. This magnitude of reduction depends on grid resolution: with a 64x64 grid the effect is about half as much as with a 32x32 grid. This is because the effect is proportional to the time step, which is about a factor of two smaller with the 64x64 grid.



Figure 2. Stabilisation of a dense layer with  $B_{DL}$ =3 and thickness 0.3 at Ra=10<sup>6</sup> and grid resolution 32x32 by the DJS algorithm. Algorithm switched off (left) or on (right).

Increasing Rayleigh number (Figure 3 centre column) results in increased Nusselt number and V<sub>rms</sub>, as expected, but the increase is lower when DJS is applied to the full density field. The resolution is 64x64 cells. The ratio (stabilised/correct) (Figure 3 bottom centre) indicates that the problem gets worse with increasing Rayleigh number. The values used here are still far below Earth's effective Rayleigh number of around 10<sup>7</sup>-10<sup>8</sup> (Schubert et al., 2001), at which the flow reduction would be much worse. This is because with higher Ra the buoyant upwellings and downwellings become narrower, but the fake advected density correction still occurs everywhere. Higher Rayleigh number solutions are not plotted here because a steady state cannot be obtained when DJS is applied to the full density field: there are oscillating boundary-layer instabilities.

# 3.3 Convection with a phase transition



In the final set of experiments (Figure 3 right column), a phase transition with zero Clapeyron slope is inserted at mid depth and its density jump is varied from 0 to 0.5, bearing in mind that the relevant jump for Earth is about 0.1. The resolution is 32x32 cells. The correct solution does not change, as the phase change density perturbation is applied in a Boussinesq-like manner, affecting the buoyancy term but nothing else. Solutions indicate that incorrectly applying DJS to the phase change density jump results in a reduction in convective vigour of a slightly lower magnitude than that obtained with a gradual density increase, but still large enough that the effect should be avoided.

Figure 3. Influence of using DJS on full density for experiments with (left column) varying density increase with depth, (middle column) varying Rayleigh number and (right column) a phase transition with different density jumps.

# 3.4 Correct functioning of DJS with all density contributions


Figure 4 shows a convection result with all density variations switched on. The air layer and deep dense layer interfaces are stable with no oscillations, and convection is not inhibited. Two-layered convection is established due to the thick dense layer. The middle row contrasts the total density field (middle left) with the compositional-only density perturbation used in the DJS algorithm (middle right).

Figure 4. A test with all density variations included (sticky air, dense layer, compressibility and a phase transition). On a 64x64 grid with Ra= $10^6$ , air thickness 0.1, dense layer thickness 0.3,  $\rho_{cmb}/\rho_{surf}=2$ ,  $\Delta\rho_{phase}=0.1$ .

# 4. Discussion and Conclusions


# 195 4.1 Isolating the composition-dependent density gradient

The results above demonstrate the importance of using only the composition-related density gradient, not the full density gradient, in the DJS algorithm. For a simple convection program like the one used here, the composition-related density gradient is straightforward to isolate. However, if the code has been written in such a manner that compositional, pressure, phase and temperature effects are combined in a single density(T,p,C) function, such as the StagYY code (Tackley, 2008), then this is more difficult. A practical solution is to perform twice as many density evaluations for each grid cell, as detailed below.

The composition-related density gradient may be calculated by subtracting the density gradient for a fixed composition from the total density gradient:

$$\left(\frac{\partial \rho}{\partial z}\right)_{C} = \left(\frac{\partial \rho}{\partial z}\right)_{total} - \left(\frac{\partial \rho}{\partial z}\right)_{fixedC}$$
(17)

Using finite differences, the total density gradient is approximated as:

$$\left(\frac{\partial \rho}{\partial z}\right)_{total} \approx \frac{1}{\Delta z} \left(\rho(T_u, C_u, z_u) - \rho(T_l, C_l, z_l)\right) \tag{18}$$

where "u" denotes the upper cell, "l" denotes the lower cell and  $\Delta z$  is the grid spacing. As the upper and lower cells can have different compositions, when calculating the density gradient for fixed composition it is best to calculate it for both compositions and average:

$$\left(\frac{\partial \rho}{\partial z}\right)_{fixedC} \approx \frac{1}{2\Delta z} \left(\rho(T_u, C_u, z_u) - \rho(T_u', C_u, z_l) + \rho(T_l', C_l, z_u) - \rho(T_l, C_l, z_l)\right) \tag{19}$$

Subtracting (19) from (18) leads to:

$$\left(\frac{\partial \rho}{\partial z}\right)_{C} \approx \frac{1}{2\Delta z} \left(\rho(T_{u}, C_{u}, z_{u}) + \rho(T_{u}', C_{u}, z_{l}) - \rho(T_{l}', C_{l}, z_{u}) - \rho(T_{l}, C_{l}, z_{l})\right) \tag{20}$$

In these expressions, primes on temperatures denote that they are extrapolated adiabatically to the required

This expression has been found to work well in recent tests of the StagYY convection code (Tackley, 2008).

locations, i.e.  $T'_u$  is  $T_u$  extrapolated adiabatically to  $z_l$ . This is because, while the focus of equation (20) is on composition, it also works on temperature, i.e. it subtracts the density gradient at fixed temperature leaving the density gradient that is due to a temperature gradient. This is appropriate, as density differences due to temperature differences are advected with the flow, but generally they are much smaller than the composition-based density gradients that are of concern here.

If one wishes to include horizontal density gradients in the DJS algorithm as in equation (4), the procedure is the same as that above (equations 17-20) for each horizontal direction.

# 4.2 Conclusions



The density-jump stabilisation algorithm of (Duretz et al., 2011) is an effective method of preventing numerical oscillations of internal compositional layers as well as of a free surface. However, it is essential that the density gradient used in the algorithm is that for compositional density variations only, otherwise severe artefacts result. If the used density gradient incorrectly includes the steady density increase with depth due to adiabatic compression and/or density jumps due to phase transitions, a severe reduction of convective vigour results. This reduction increases with Rayleigh number but decreases with increasing numerical resolution. Isolating the compositional component of the density gradient can be straightforwardly done using the approach presented in this paper.

Code availability. The exact version of the Julia code used to produce the results and figures in this paper is archived on Zenodo under the MIT license under DOI 10.5281/zenodo.15115816 (Tackley and ETH Zurich, 2025). No input data or additional scripts are required. The Julia script used to produce the graphs in Fig. 3 is also archived there.

**Competing interests.** The author declares that he has no conflict of interest.

**Special issue statement**. Advances in numerical modelling of geological processes.

Acknowledgments. Identifying and fixing these problems at this point in time was motivated by dense layer oscillations identified by Elena Zaharia and Maxim Ballmer. Helpful reviews were provided by Mingming Li and Boris Kaus. Albert de Montserrat Navarro helped the author to improve the plotting in the final version.

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
