# Peer review of "On stabilisation of compositional density jumps in compressible mantle convection simulations"

_EGUsphere, 2025_

## Author Response (AR1)

**Response to referee comments**

Reviewer comments are pasted in dark grey, and responses are in red.

-----Referee #1: Mingming Li

This study solves an important numerical artefact when simulating mantle convection with significant density contrast between layers. The paper is well written and easy to follow. The results are supported by the experiments. I recommend 'accept' after a few very minor changes as detailed below.

Thank you for your suggestions, which I have followed.

Line 60, 118: 'Where' -> 'where'

**Corrected**

Line 108: \Theta is not defined in Eq. (13), or change \Theta to 'the \theta in Equation (5)'

**Changed to " $\Theta$ in equation (5)"**

Equation 16: I do not quite understand this equation. Could you include some derivation for this equation?

I have added an extra step in equation (16) and some explanation below it.

$$"\Delta\rho_C = C_{air}\Delta\rho_{air} + C_{DL}\Delta\rho_{DL} = -C_{air} - C_{DL}B_{DL} / B_{air}$$
(16)

(noting that  $\Delta \rho_{air} = -1$  nondimensional and  $\Delta \rho_{DL} = \Delta \rho_{air} B_{DL} / B_{air} = -B_{DL} / B_{air}$ ),

-----Referee #2: Boris Kaus

This is a very interesting manuscript that points out that the stabilisation algorithm as originally proposed in Kaus et al. (2010) requires modifications before it can be applied to cases with adiabatic compression. It is well-written and logically argued, and the provided julia code works well.

I have only a few very minor suggestions (listed below), but would strongly suggest acceptance as soon once this is accounted for.

**Thank you for your suggestions, which I have followed.**

I. 25: Our original algorithm was in fact applied everywhere in the domain and not just at cells with a free surface. Obviously, since those cells have the largest density difference, most of the correction ended up being applied here.

**Clarified in the revised manuscript.**

I. 25: An additional paper that proposed a slightly different algorithm along with rigorous numerical stability analysis was by Rose et al. (2017), which is at least worth citing in this context. Also, fully implicit timestepping schemes can be used to circumvent the problem altogether as discussed by Popov and Sobolev (2008) and Kramer et al. (2012)

Added sentences about these references to the revised manuscript.

I. 147: Nusselt numbers are the identical => Nusselt numbers are identical (remove "the")

**Corrected**

I. 147: the rms. velocity => the rms velocity (remove ".")

**Changed**

I. 155: DGS => DJS

**Corrected**

**References**

- Kramer, S.C., Wilson, C.R., Davies, D.R., 2012. An implicit free surface algorithm for geodynamical simulations. Physics of the Earth and Planetary Interiors 194–195, 25–37. https://doi.org/10.1016/j.pepi.2012.01.001
- Popov, A.A., Sobolev, S.V., 2008. SLIM3D: A tool for three-dimensional thermomechanical modeling of lithospheric deformation with elasto-visco-plastic rheology. Physics of the Earth and Planetary Interiors 171, 55–75. https://doi.org/10.1016/j.pepi.2008.03.007
- Rose, I., Buffett, B., Heister, T., 2017. Stability and accuracy of free surface time integration in viscous flows. Physics of the Earth and Planetary Interiors 262, 90–100. https://doi.org/10.1016/j.pepi.2016.11.007

---

## Author Response (AR2)

**Response to Topic Editor's comments**

The technical corrections I suggest relate to enhancing the figures, improving their layout, potentially homogenising font sizes and reducing, when possible, spacing between subplot panels.

All figures have been remade using Julia's GLMakie package instead of Plots (as in the original version), so now look much better.

This also required posting a new version of the software (1.1) on Zenodo.